# Optimized 3D-Printed Polylactic Acid/Graphene Oxide Scaffolds for Enhanced Bone Regeneration

**DOI:** 10.3390/bioengineering12111192

**Published:** 2025-11-01

**Authors:** Jung-Tae Lee, Dajung Lee, Ye-Seul Jung, Sung-Ho Lee, Sungtae Kim, Bongju Kim, Dong-Wook Han

**Affiliations:** 1Department of Periodontics, One-Stop Specialty Center, Seoul National University, Dental Hospital, Seoul 03080, Republic of Korea; jungtae1308@hanmail.net; 2Implant R&D Center, Osstem Implant Co., Ltd., Seoul 07789, Republic of Korea; dajueng90@gmail.com; 3Dental Life Science Research Institute, Seoul National University, Dental Hospital, Seoul 03080, Republic of Korea; yeesul333@gmail.com (Y.-S.J.); shlee79@snu.ac.kr (S.-H.L.); 4Department of Periodontology, Dental Research Institute, Seoul National University School of Dentistry, Seoul 03080, Republic of Korea; kst72@snu.ac.kr; 5Department of Cogno-Mechatronics Engineering, Pusan National University, Busan 46241, Republic of Korea

**Keywords:** 3D printing, polylactic acid, graphene oxide, scaffold, bone regeneration, pore size, biomechanics

## Abstract

Background: Three-dimensional (3D) printed scaffolds have emerged as promising tools for bone regeneration, but the optimal structural design and pore size remain unclear. Polylactic acid (PLA) reinforced with graphene oxide (GO) offers enhanced mechanical and biological performance, yet systematic evaluation of architecture and pore size is limited. Methods: Two scaffold architectures (lattice-type and dode-type) with multiple pore sizes were fabricated using UV-curable PLA/GO resin. Physical accuracy, porosity, and mechanical properties were assessed through compression and fatigue testing. Based on in vitro screening, four pore sizes (930 μm, 690 μm, 558 μm, 562 μm) within the dode-type structure were analyzed. The 558 μm and 562 μm scaffolds, showing distinct fracture thresholds, were further evaluated in rat and rabbit calvarial defect models for inflammation and bone regeneration. Results: In vitro testing revealed that while 930 μm and 690 μm scaffolds exhibited superior compressive strength, the 562 μm scaffold showed a unique critical fracture behavior, and the 558 μm scaffold offered comparable stability with higher resistance to premature failure. In vivo studies confirmed excellent biocompatibility in both groups, with early bone formation favored in the 558 μm scaffold and more continuous and mature bone observed in the 562 μm scaffold at later stages. Conclusions: This stepwise strategy—from structural design to pore size screening and preclinical validation—demonstrates that threshold-level mechanical properties can influence osteogenesis. PLA/GO scaffolds optimized at 558 μm and 562 μm provide a translationally relevant balance between mechanical stability and biological performance for bone tissue engineering.

## 1. Introduction

Bone defects resulting from trauma, disease, or congenital abnormalities present significant clinical challenges, necessitating the development of effective bone regeneration strategies to restore structural integrity and biological function. Scaffolds have emerged as critical components in bone tissue engineering, serving as temporary three-dimensional frameworks that provide essential structural support while mimicking the natural extracellular matrix environment [1,2]. These bioengineered constructs facilitate crucial cellular processes including cell adhesion, proliferation, and differentiation, ultimately guiding the formation of new bone tissue [3,4]. The advent of three-dimensional printing technology has revolutionized scaffold fabrication by enabling precise control over architectural features such as pore size, porosity distribution, and mechanical properties, allowing for patient-specific designs tailored to individual bone repair requirements [5,6].

Despite their potential, current scaffold technologies face several significant limitations that hinder optimal bone regeneration outcomes. Common challenges include insufficient mechanical strength to withstand physiological loads, poor osteoconductivity that limits bone-forming cell attachment and growth, inadequate bioactivity for promoting cellular responses, and suboptimal degradation rates that may not align with natural bone healing timelines [7,8]. Among the promising biomaterials for scaffold construction, polylactic acid (PLA) has gained considerable attention due to its excellent biocompatibility, biodegradability, and processability, while graphene oxide (GO) has emerged as a valuable reinforcing agent capable of enhancing both mechanical and biological properties of composite scaffolds [9,10].

PLA, while biocompatible, exhibits inherent drawbacks including brittleness under mechanical stress and hydrophobic surface properties that can impede cell adhesion and proliferation [11,12]. The incorporation of GO aims to address these limitations by improving mechanical reinforcement, enhancing surface hydrophilicity, and providing bioactive sites for cellular interaction. Recent studies have demonstrated that GO-PLA composites exhibit improved biocompatibility and mechanical properties compared to pure PLA scaffolds, with 3D-printed PLA/GO membranes successfully promoting guided bone regeneration through enhanced osteoblast proliferation [5,9]. The synergistic combination of these materials leverages GO’s ability to improve mechanical strength while maintaining PLA’s biodegradability, creating scaffolds with superior performance for bone tissue engineering applications. However, challenges remain regarding uniform dispersion within the polymer matrix and potential cytotoxicity concerns at higher concentrations [13,14,15]. Furthermore, 3D printing technology enables precise control over scaffold geometry, porosity, and internal architecture, allowing for the fabrication of patient-specific implants tailored for bone regeneration. Leveraging these technological advantages, systematic and scientific optimization of scaffold design and composition is essential to enhance both the biological and mechanical performance of 3D-printed composite scaffolds and to advance their clinical translation [16,17].

Despite these promising developments, significant gaps remain in the current literature regarding the systematic optimization of PLA/GO composite scaffolds for bone regeneration. While successful bone regeneration has been demonstrated using 3D printed GO-loaded polymer scaffolds through histomorphometric evaluation, these studies were limited to single polymer systems without comprehensive parameter optimization [9,18,19]. Additionally, functionalization of PLA/GO scaffolds with bioactive compounds has been shown to enhance osteogenesis, yet this approach focused primarily on surface modification rather than systematic printing parameter optimization [20,21]. Similarly, GO incorporation as a coupling agent in PLA-based nanocomposite scaffolds has achieved improved mechanical properties and bioactivity, but lacked comprehensive evaluation of 3D printing parameters’ effects on scaffold performance [10]. These studies collectively highlight the potential of PLA/GO composites while revealing the need for systematic optimization of 3D printing parameters, including pore size and porosity structure to achieve maximum scaffold performance for clinical applications. Therefore, the purpose of this study was to fabricate 3D-printed PLA/GO composite scaffolds, identify the optimized structure through physical and mechanical characterization, and verify their safety and efficacy through in vivo animal testing. The novelty of this work lies in three key aspects: (1) the establishment of an integrated design-to-animal model workflow that connects design, 3D printing, mechanical evaluation, and animal experimentation; (2) the precise optimization of pore size (558–562 μm) to systematically investigate how structural variations influence mechanical durability and bone regeneration capacity; and (3) the introduction of a quantitative structure–property relationship between scaffold geometry and compressive behavior, thereby addressing the limitations of previous PLA/GO scaffold studies.

## 2. Materials and Methods

### 2.1. Materials

The experimental design of this study is schematically illustrated in Figure 1.

The 3D printing resin scaffolds used in this study were provided by Luvantix ADM Co., Ltd. (Gukjegwahak 8-ro, Yuseong-gu, Daejeon, Republic of Korea). This UV-curable resin was specifically formulated to exhibit tailored physicochemical and biological properties and consisted of dendritic PLA–PCL oligomers, zwitterionic monomers, and graphene [6]. During the fabrication of the GO–PCL composite material, the concentration of leachates was strictly maintained below 250 µg/mL, and a uniform coating and dispersion were achieved using the Meniscus Dragging Deposition technique. Preliminary analyses, including SEM, Raman spectroscopy, XPS, and contact-angle measurements, were conducted to evaluate surface morphology, chemical composition, and wettability. STL files of the 3D scaffold models were designed using CHITUBOX V1.9.0 software (Phrozen, Hsinchu City, Taiwan), and the GO–PLA precursor solution was loaded into an LCD/DLP-based photopolymerization 3D printing system (Phrozen Sonic Mini 8K, Hsinchu City, Taiwan) optimized for high-resolution resin processing. The UV-curable PLA/GO composite resin exhibited a viscosity of 1700–4000 cPs, allowing for stable layer-by-layer fabrication under 405 nm UV exposure with an optimized exposure time of 5–8.5 s per layer at an intensity of 2.0 mW/cm^2^. The printing accuracy was maintained within ±1–4.7%, and the designed pore architectures (50–300 µm) were precisely reproduced with negligible structural deformation. Because the LCD/DLP system operates through optical curing rather than thermal extrusion, parameters such as printing temperature and nozzle diameter are not applicable. These optimized printing conditions ensured high dimensional precision, uniform porosity, and reproducible scaffold fabrication suitable for both mechanical characterization and in vivo validation.

To identify the optimal scaffold architecture, a series of hypothetical porous structural designs were developed with varied pore sizes and corresponding porosities. Two distinct unit cell structures were modeled: Structure 1 (lattice-type) with pore sizes of 558 μm (44.01%), 652 μm (56.36%), 752 μm (65.88%), 875 μm (73.35%), and 1000 μm (78.40%); and Structure 2 (dode-type) with pore sizes of 750 μm (32.87%), 875 μm (50.78%), 1000 μm (58.39%), 1125 μm (66.11%), and 1250 μm (72.27%). These structures served as experimental scaffold candidates for evaluating print accuracy, geometric fidelity, mechanical strength, and biomechanical performance. The results of these assessments informed the selection of optimal scaffolds for subsequent in vivo animal experiments.

### 2.2. Physical and Mechanical Property Analysis of Materials

#### 2.2.1. Pore Size and Printing Accuracy

The printed scaffolds were analyzed to assess dimensional accuracy by comparing the measured pore size of the printed structure with the designed CAD reference model. After fabrication, representative samples from each group were examined using a stereo microscope (Leica M205 C, Leica Microsystems, Wetzlar, Germany) at 30× magnification. High-resolution images were captured, and pore diameters were measured using Image J v1.54 (National Institutes of Health, Bethesda, MD, USA). For each scaffold, five randomly selected pores were measured, and the average value was used for analysis. Calibration was performed using a stage micrometer prior to measurement. The percentage error between the printed pore size and the designed model was calculated to evaluate printing accuracy.

#### 2.2.2. Biomechanical Analysis

Considering the 10 scaffold designs, models were constructed using a 3D CAD program and meshes were formed through the HyperWorks (Altair Engineering Inc., Troy, MI, USA) program. All scaffold structures were meshed with identical mesh sizes. Since dental scaffolds and medical scaffolds are primarily subjected to compressive forces, loads were applied in the compression direction. The analysis procedure involved constructing a rigid plate at the upper portion of the scaffold and forming a support structure based on a rigid plate at the lower portion of the scaffold. The lower portion was set to prevent displacement and rotation in all directions, while the interfaces between the scaffold and the plates located at the upper and lower portions were applied with a friction coefficient of 0.3. Using the loading plate at the upper portion, compressive force was applied with constant displacement in the −Z axis direction, and analysis was performed until the scaffold reached 10% strain.

#### 2.2.3. Compression Test

Compression testing was conducted in accordance with ASTM D695 international standard [22]. Test specimens were prepared in either prism or cylindrical form with dimensions of 12.7 × 12.7 × 25.4 mm^3^ and a minimum of five specimens were used for the evaluation. Prior to testing, the gauge length section of each specimen was measured using a vernier caliper to record both width and height. All tests were performed under controlled environmental conditions of 23 ± 2 °C and 50 ± 10% relative humidity. A universal testing machine capable of recording load–displacement data was used to evaluate compressive strength and strain, with a constant crosshead speed of 1.3 mm/min applied [23]. The test was terminated either upon specimen fracture or at the point where a decrease in load was observed on the load–displacement curve.

#### 2.2.4. Fatigue Test

Fatigue tests were performed to evaluate the cyclic compressive durability of the printed scaffolds. The procedure was based on the ASTM D695 standard for compressive testing of rigid plastics, with modifications to apply cyclic loading conditions similar to those reported in previous scaffold fatigue studies [10,15,23]. Specimens (12.7 × 12.7 × 25.4 mm^3^) were subjected to sinusoidal cyclic compressive loads at a frequency of 10 Hz, with amplitudes ranging from −30 to −300 N and −70 to −700 N.

The selected frequency of 10 Hz was chosen to approximate the physiological cyclic loading environment of masticatory forces while enabling accelerated assessment of scaffold durability. Similar frequency ranges (1–15 Hz) have been widely used in previous fatigue studies on polymeric and composite scaffolds to simulate in vivo mechanical conditions and reduce testing time. Negative load values indicate compression in the testing system. Each condition was tested up to 5 × 10^6^ cycles or until failure. The mechanical data were analyzed in terms of compressive stress amplitude and number of cycles to failure.

### 2.3. In Vivo Animal Study

#### 2.3.1. Preparation of Animals

Animal experiments were approved by the Institutional Animal Care and Use Committee (IACUC) of Seoul National University (IACUC No. SNU-221130-1-1). Sprague Dawley rats (150–200 g, Orient Bio, Gapyeong, Republic of Korea) and Male New Zealand white rabbits (2–3 kg; KNOTUS, Incheon, Republic of Korea) were used. The animals were housed in a Specific Pathogen-Free (SPF) environment with controlled conditions: temperature at 21 ± 1 °C, relative humidity at 55%, and a 12 h light/dark cycle (lights on: 07:30–20:00; lights off: 20:00–07:30). They were fed a standard diet (Purina Rodent Chow, Purina Co., Ltd., Seoul, Republic of Korea) and monitored for health and body weight. All procedures were conducted in accordance with the Korean Guide for the Use and Care of Laboratory Animals and IACUC regulations.

#### 2.3.2. Sprague Dawley Rat Calvarial Defect Model

This rat experiment was conducted to evaluate the inflammatory response. After 1 week of the quarantine period, the rats were anesthetized by intraperitoneal injection using a 3 mL mixture (100 mg/kg) of pentobarbital (Hanlim Pharm, Co., Ltd., Gyeonggi, Republic of Korea) and chloral hydrate (Sigma-Aldrich. Co., Ltd., Oakville, ON, Canada). Two 5 mm circular calvarial defects were created bilaterally on either side of the sagittal suture in Sprague Dawley rats (n = 12). Scaffolds with pore sizes of 558 μm and 562 μm were implanted into each defect. At 1, 2, and 4-week, post-implantation, animals (four rats each week) were sacrificed, and histological analyses (Hematoxylin & Eosin and Masson’s trichrome staining) were conducted.

#### 2.3.3. New Zealand White Rabbit Calvarial Defect Model

This rabbit experiment (n = 8) was conducted following the above rat study to evaluate the inflammatory response and new bone formation in a higher animal model. Ketamine (35 mg/kg) and xylazine (5 mg/kg) were injected intramuscularly for anesthesia induction. Intravenous levofloxacin (5 mg/kg) and warm normal saline solution (2 mL/kg/h) were infused to prevent infection and dehydration. After applying iodine solution to the surgical site, 0.9 mL of 2% lidocaine with epinephrine was injected subcutaneously. A 5 mm calvarial defect was created in New Zealand white rabbits, and scaffolds with pore sizes of 558 μm and 562 μm were implanted. At 4 and 8-week, post-implantation, animals were sacrificed. Histological analysis was performed using hematoxylin and eosin (H&E) and Masson’s Trichrome staining to evaluate general morphology and collagen deposition. The Masson’s Trichrome method was conducted according to the standard procedure described by Bancroft and Gamble [23], where mineralized tissue appears red, collagen fibers blue, and cytoplasm light red.

The difference in the number of calvarial defects between rats and rabbits was determined based on anatomical and experimental considerations. In the rat model, two 5 mm bilateral defects were created to allow for within-subject comparison of the two scaffold types (558 μm and 562 μm) while minimizing the total number of animals used. In contrast, the rabbit calvarium provides a larger surface area, and a single standardized 5 mm defect was sufficient for independent evaluation of bone regeneration. Creating multiple defects in rabbits was avoided to prevent possible overlap of healing zones and to adhere to ethical principles of minimizing surgical trauma.

### 2.4. Statistical Analysis

All data were analyzed using GraphPad Prism 10.0 (GraphPad Software, Boston, MA, USA) and expressed as mean ± standard deviation (SD). One-way analysis of variance (ANOVA) followed by Tukey’s post hoc test was used to determine statistical significance among multiple groups. This combination was chosen because it is appropriate for parametric data with normal distribution and equal variance, allowing for reliable pairwise comparisons while controlling Type I error. Normality and homogeneity of variance were verified before performing ANOVA. A *p*-value < 0.05 was considered statistically significant [24,25,26].

## 3. Results

### 3.1. Physical and Mechanical Properties

#### 3.1.1. Pore Size and Printing Accuracy

To evaluate the shape accuracy of the output compared to the reference model, porosity errors were analyzed as shown in Table 1 [27]. The porosity of structure 1 was measured at 49.01%, 53.3%, 59.38%, 73.33%, and 76.59% with errors relative to the reference model of 5%, −3.06%, −6.5%, −0.01%, and −1.81%, respectively. The porosity of structure 2 was measured at 33.01%, 51.04%, 57.7%, 66.07%, and 65.53%, with errors relative to the reference model of 0.14%, 0.26%, −0.68%, −0.04%, and −6.74%, respectively. Structure 1 exhibited the largest shape errors at pore sizes of 0.55 mm and 0.75 mm, while Structure 2 showed the highest error at a pore size of 1250 μm. Structure 1 demonstrated an average error of 3.28%, whereas Structure 2 showed an average error of 1.57%.

The dimensional error for each scaffold was calculated as the percentage difference between the measured and designed pore sizes or porosities according to the following equation: (Measured − Designed)/Designed × 100. Positive values represent overestimation and negative values underestimation of the designed pore dimension. The reported average error for each structure was obtained as the mean of the absolute percentage errors across all pore sizes, which provides a balanced indication of overall printing accuracy without cancellation of positive and negative deviations.

#### 3.1.2. Biomechanical Analysis—Compression Testing

The compressive strength of the two scaffold architectures was inversely related to porosity, with higher porosity generally resulting in reduced mechanical strength. In Structure 1, compressive strength remained relatively consistent at approximately 3 MPa across porosities above 73%. In contrast, Structure 2 exhibited stable compressive strength values when the porosity exceeded 66%. Given that a compressive strength above 3 MPa is considered sufficient to provide mechanical support in dental applications, all five porosity levels evaluated in this study demonstrated adequate mechanical properties for clinical use. However, considering dynamic loading conditions such as temporomandibular joint movement and peak occlusal forces, an optimal pore size of 750 μm (porosity 65.38%) for Structure 1 and 875 μm (porosity 50.78%) for Structure 2 is suggested to ensure both structural integrity and functional performance (Figure 2A,B).

The stress distribution across the ten scaffold designs revealed distinct patterns depending on the scaffold structure. Structure 1 primarily transmitted stress in the direction perpendicular to the applied load. Notably, when the porosity exceeded 70%, bending deformation was observed at both ends and the lower portion of the scaffold under 10% strain, resulting in a loss of load-bearing capacity. In contrast, Structure 2 demonstrated stress transmission through interconnected circular features within the porous framework when subjected to perpendicular loading. The internal porous architecture of Structure 2 consisted of circular units from which eight columns extended, with stress concentration occurring at the junctions between the circular bases and the columns. Overall, Structure 2 exhibited more effective stress dispersion compared to Structure 1 (Figure 2C,D).

#### 3.1.3. Compression Test

Through scaffold architecture screening, it was confirmed that Structure 2 (dode type) was more advantageous as a scaffold compared to Structure 1. To further determine the optimal pore size within the Structure 2 design, four scaffolds with different pore sizes (930 μm, 690 μm, 558 μm, and 562 μm) were fabricated.

As shown in Table 2, the compression test results revealed yield loads of 400–1290 N, yield strains of approximately 3.5–4.5%, yield strengths of 2.8–6.72 MPa, and elastic moduli of 150–250 MPa. The force acting on teeth during mastication varies depending on the type of food being chewed, but when chewing biscuits, carrots, and cooked meat, the force acting on teeth ranges from 190–260 N, and the maximum masticatory force that can be applied to molars is known to be 500–700 N [28]. The 690 μm group exhibited the highest yield load (1107.06 ± 114.56 N), yield strength (6.72 ± 0.68 MPa), and elastic modulus (230.20 ± 25.08 MPa), showing significant improvement compared to the 930 μm group (*p* < 0.05). In contrast, the 562 μm group demonstrated the lowest values across most parameters, particularly a reduced yield load (432.34 ± 53.23 N) and elastic modulus (144.22 ± 22.70 MPa), both significantly lower than those of the 690 μm group (*p* < 0.05). Interestingly, the 558 μm group showed mechanical properties comparable to or slightly better than the 930 μm group, with significant differences against the 690 μm and 562 μm groups in several parameters. Its yield displacement (1.04 ± 0.11 mm) and yield strain (4.23 ± 0.45%) were significantly higher than those of the 562 μm group, indicating improved deformability and resistance to fracture under compressive load. After compression testing, fracture was confirmed to occur at strain levels above 5% in the 562 μm and 558 μm specimens, while the 930 μm and 690 μm specimens showed no significant fracture up to approximately 15% strain (Appendix A).

#### 3.1.4. Fatigue Test

Table 3 presents the results of fatigue tests conducted on porous scaffold materials for dental applications. Specimens with pore sizes of 930 μm, 690 μm, 562 μm, and 558 μm were subjected to cyclic compressive loading. Under a loading condition of −30 to −300 N, all specimens withstood 5,000,000 cycles without failure. However, when the applied load was increased to −70 to −700 N, specimens with pore sizes of 562 μm, and 558 μm exhibited structural failure. These findings suggest that scaffolds with larger pore sizes (930 μm and 690 μm,) maintained structural integrity even under high cyclic loads, while those with smaller pores (562 μm, and 558 μm) were susceptible to fatigue failure beyond 300 N. The scaffold materials showed no failure under 300 N, which is within the range of normal chewing forces, indicating that they have adequate mechanical stability for dental use.

### 3.2. In Vivo Animal Study

#### 3.2.1. Evaluation of PLA/GO Scaffold in a Rat Preclinical Model

Figure 3 illustrates the results of the in vivo experiment conducted in rats. The initial H&E-stained images revealed the presence of inflammatory tissue surrounding the implanted scaffolds, characterized by infiltration of inflammatory cells including osteoblasts and osteoclasts. At week 1, a considerable presence of blood and inflammatory cells was observed within the scaffold pores in both the 558 μm and 562 μm scaffold-implanted groups. By 2 weeks, a marked reduction in inflammatory cells was observed, consistent with resolution of acute inflammation and progression toward tissue remodeling. By week 4, no inflammatory cells were detected in either group, and the newly formed bone, which first appeared at week 2, had expanded and occupied a broader area within the scaffold structure. Masson’s Trichrome staining demonstrated time-dependent new bone matrix development within scaffold pores. At 1-week, minimal osteoid formation (blue staining) was seen in both groups. However, by 2 weeks, there was a noticeable increase in collagen-rich osteoid, especially in the 558 μm group. At 4 weeks, both groups exhibited mature lamellar bone formation, with the 562 μm group showing more continuous and dense bone matrix. The experimental period (1, 2, and 4 weeks) showed a statistically significant effect on bone formation (*p* < 0.0001). In contrast, pore size (558 µm vs. 562 µm) did not exhibit a significant effect (*p* = 0.6153), and no significant interaction between the two factors was observed (*p* = 0.1367). These results suggest that while early-stage osteogenesis was slightly favored in smaller pores, the larger 562 μm pores supported more extensive bone maturation over time.

#### 3.2.2. Evaluation of PLA/GO Scaffold in a Rabbit Preclinical Model

Scaffolds with two different pore sizes (558 μm and 562 μm) were implanted into rabbit calvarial defects and evaluated histologically at 4 and 8-week post-implantation. Masson’s Trichrome (MT) staining was used to assess new bone formation within the pores of the 3D-printed scaffolds, while H&E staining was employed to evaluate inflammation. At 4 weeks, both scaffold types exhibited new bone formation within the pore structures. The 558 μm group showed 21.25 ± 3.44% new bone area, while the 562 μm group demonstrated 20.00 ± 3.03%. By 8 weeks, the amount of new bone had increased slightly in both groups, with 24.66 ± 4.90% in the 558 μm group and 23.95 ± 1.01% in the 562 μm group. Bone formation significantly differed according to the experimental period (4 weeks vs. 8 weeks, *p* = 0.0077), whereas the effects of pore size and its interaction were not statistically significant (*p* = 0.8356). Importantly, H&E staining confirmed the absence of inflammatory cell infiltration within the scaffold pores at all time points, indicating good biocompatibility of the implanted materials. These results demonstrate that both pore sizes supported new bone formation over time, with comparable osteoconductive performance and no histological signs of inflammation (Figure 4).

## 4. Discussion

The selection of scaffold structures and pore sizes in this study followed a stepwise and rational process. Two distinct scaffold architectures (Structure 1: lattice-type and Structure 2: dode-type) were initially compared to evaluate how geometry affects stress distribution and mechanical stability. Structure 2 demonstrated more uniform stress dispersion and higher mechanical resilience, which justified its selection for subsequent optimization. Within this optimized architecture, four different pore sizes (930 μm, 690 μm, 558 μm, and 562 μm) were tested in vitro. While larger pores (930 μm and 690 μm) showed favorable mechanical strength, they were less advantageous for initial cell adhesion and bone ingrowth. Notably, the 562 μm scaffold exhibited a unique “critical fracture” behavior, with microcracks appearing at 3–4% strain and complete failure beyond 5–6% strain, whereas the 558 μm scaffold demonstrated nearly identical characteristics but with greater resistance to premature fracture. For this reason, these two pore sizes were intentionally selected for in vivo validation. This design allowed us to directly test whether subtle but functionally critical differences in mechanical stability would translate into distinct biological outcomes. Indeed, our in vivo results demonstrated that while both groups supported bone formation, the 562 μm scaffold showed more mature and continuous bone formation at later stages, highlighting that threshold-level mechanical properties may influence long-term bone regeneration. Unlike previous studies [5,9,10], which primarily examined GO concentration or general scaffold morphology, this study therefore provides a systematic and translational framework, progressing from structural design, through pore size optimization, to in vivo validation. The deliberate choice of 558 μm and 562 μm scaffolds underscores that scaffold design is not only about achieving mechanical safety but also about exploring critical thresholds that may shape biological outcomes.

The optimized printing parameters resulted in scaffolds with increased compressive strength, better surface roughness conducive to cell adhesion, and improved interconnected porosity that facilitated nutrient transport and waste removal. Furthermore, in vitro studies revealed enhanced osteoblast proliferation, differentiation, and mineralization on the optimized PLA/GO scaffolds, suggesting their potential for successful bone tissue engineering applications [27,29]. The improved performance of optimized PLA/GO scaffolds can be attributed to several synergistic mechanisms. The systematic optimization of printing parameters, including layer height, printing speed, and infill density, likely resulted in better polymer chain alignment and reduced internal stresses, leading to enhanced mechanical properties. The incorporation of graphene oxide at optimal concentrations provided nucleation sites for polymer crystallization while maintaining the biodegradable nature of PLA [10,30]. Additionally, the controlled porosity achieved through parameter optimization created an ideal microenvironment that mimicked the natural bone extracellular matrix, promoting cell infiltration and vascularization. The enhanced surface area provided by GO incorporation may have facilitated protein adsorption and subsequent cellular interactions, ultimately contributing to improved osteogenic responses [8,31,32].

Several studies in the literature support our findings regarding the beneficial effects of optimized PLA/GO composite scaffolds. Belaid et al. demonstrated that 3D-printed PLA/graphene oxide scaffolds have enhanced mechanical properties and promote bone cell proliferation and mineralization compared to pure PLA scaffolds [5]. Similarly, Jang et al. reported that 3D-printed PLA/graphene oxide scaffolds enhance bone regeneration [9]. Tavakoli et al. showed that 3D-printed PLA/graphene oxide/hardystonite nanocomposite scaffolds exhibit enhanced mechanical properties and bioactivity for bone tissue regeneration [10]. Furthermore, Mashhadi Keshtiban et al. confirmed that graphene oxide surface treatment of 3D-printed polylactic acid scaffolds enhances bone regeneration [11], supporting our systematic approach to parameter optimization.

However, some studies present contrasting findings to our results. Alazab et al. reported that 3D-printed PCL scaffolds with 1 wt% graphene oxide enhanced bone regeneration compared to pure PCL or 3 wt% graphene oxide scaffolds, suggesting that higher GO concentrations may not always be beneficial [18]. This finding indicates that concentration-dependent effects may limit the beneficial outcomes and that excessive GO incorporation can lead to decreased performance. Additionally, some investigations have indicated that over-optimization of certain printing parameters may result in compromised scaffold integrity or reduced printability. These conflicting findings highlight the importance of balanced optimization and the need for comprehensive evaluation of multiple parameters simultaneously rather than individual component optimization [33,34,35].

This study addresses a significant gap in the current literature by providing a systematic approach to PLA/GO scaffold optimization specifically for bone regeneration applications. While previous research has explored individual aspects of PLA/GO composites or isolated printing parameters, comprehensive optimization studies remain limited. Sánchez-Cepeda et al. demonstrated that 3D-printed PLA/GO/TCP scaffolds functionalized with poly-l-lysine exhibit enhanced structural integrity and osteogenic properties [20], but their work focused primarily on surface modification rather than systematic printing parameter optimization. Similarly, A previous study showed that polydopamine-reduced graphene oxide reinforced 3D printed PLA scaffolds exhibit enhanced properties for bone tissue engineering [8], yet this approach focused on material modification rather than processing optimization. The systematic evaluation of multiple printing parameters and their synergistic effects on scaffold performance represents a crucial advancement in the field. This research provides valuable insights into the complex relationships between material composition, processing parameters, and biological outcomes, offering a framework for future scaffold development.

The strengths of this study include the systematic approach to parameter optimization, comprehensive evaluation of both mechanical and biological properties. The multi-factorial optimization design allowed for the identification of optimal parameter combinations rather than individual variable effects. Furthermore, this study demonstrates a direct and rational progression from computational modeling and mechanical testing to biological validation. The selection of pore sizes (558 μm and 562 μm) used in the animal experiments was not arbitrary but based on systematic screening of scaffold geometries with varying porosity levels and mechanical properties. Compression and fatigue testing revealed that these dimensions offered a critical balance between structural support under masticatory forces and manufacturability within the limits of 3D printing resolution. The subsequent in vivo outcomes including reduced inflammation and increased new bone formation further validated the functional suitability of the selected pore sizes. This continuity from design optimization to biological efficacy underscores the translational potential of the scaffold fabrication strategy.

Recent results from a government-funded research project (MOTIE, Sejong-si, Republic of Korea) [36] confirmed that PLA/PCL/GO composites exhibit excellent biocompatibility, with no cytotoxic or immunogenic response at GO concentrations below 250 µg/mL. In vitro experiments demonstrated enhanced preosteoblast adhesion and proliferation, supporting the optimized GO content used in this study. The dendrimer-type PLA–PCL copolymer matrix containing zwitterionic oligomers showed gradual and synchronized degradation with bone remodeling, maintaining mechanical integrity during the healing process. The developed PLA/GO resin also achieved high-resolution 3D printing precision (±1%). It was also verified to comply with ISO 10993-5 biological evaluation standard [37]. Collectively, these findings indicate that the optimized PLA/GO scaffolds provide an ideal balance of long-term stability, biological safety, and clinical manufacturability. Furthermore, these results highlight a feasible pathway for clinical translation, including considerations of large-scale production, reproducible quality control, and regulatory compliance for patient-specific scaffold fabrication.

However, several limitations must be acknowledged. This study was conducted primarily under controlled laboratory conditions, which may not fully represent the complex in vivo environment. The evaluation period was relatively short-term, and long-term biodegradation and biocompatibility studies are needed. Additionally, this study focused on a specific cell line, and validation with primary cells and animal models would strengthen the clinical relevance of the findings. The optimization was limited to specific printing parameters, and other factors such as post-processing treatments or additional bioactive compounds were not explored.

## 5. Conclusions

This study systematically optimized 3D-printed PLA/graphene oxide scaffolds for bone regeneration. Two different architectures were initially compared, and the dode-shaped design (Structure 2) demonstrated superior stress dispersion and mechanical stability, leading to its selection for further analysis. Within this architecture, four pore sizes (930 μm, 690 μm, 558 μm, and 562 μm) were screened in vitro. While larger pores showed high mechanical strength, the 562 μm scaffold exhibited critical fracture behavior, and the 558 μm scaffold showed comparable stability with greater resistance to premature failure. These two pore sizes were deliberately selected for in vivo validation. Animal experiments confirmed biocompatibility and progressive bone formation in both groups, with the 562 μm scaffold supporting more continuous and mature bone at later stages. This stepwise approach—from structural comparison through pore size screening to preclinical validation—underscores the translational potential of PLA/GO scaffolds and highlights the importance of threshold-level mechanical properties in modulating bone regeneration.

## Figures and Tables

**Figure 1 bioengineering-12-01192-f001:**
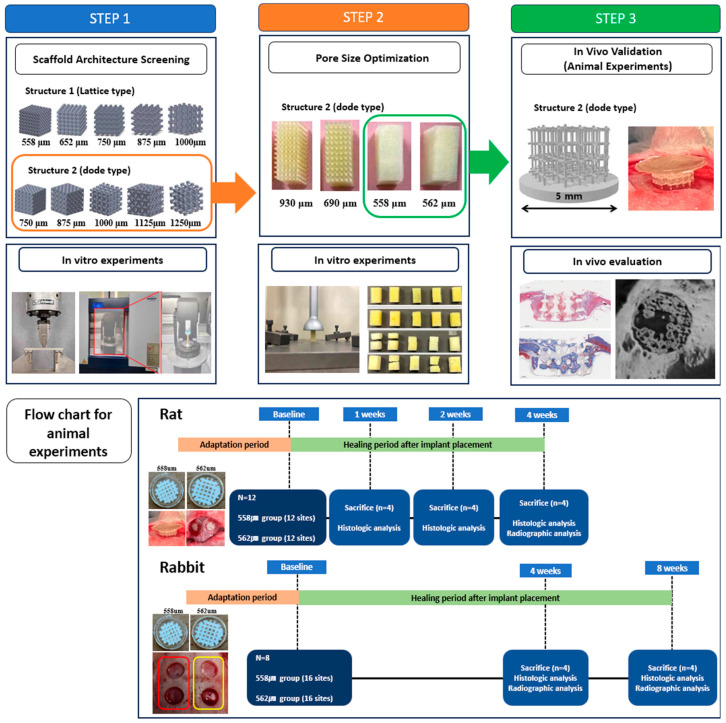
The workflow consisted of fabricating optimized PLA scaffolds reinforced with graphene oxide (GO) via 3D printing and subsequently applying the selected scaffolds in in vivo experiments. The identification of the ideal scaffold was divided into three steps. In the first step, two types of scaffold architectures (Structure 1 and Structure 2), each with five different pore sizes, were fabricated and evaluated in vitro. In the second step, additional in vitro experiments with four different pore sizes were performed within the selected architecture (structure 2) to determine pore sizes suitable for preclinical evaluation. Finally, two scaffolds with distinct pore sizes were selected and implanted into rat and rabbit calvarial defect models for in vivo validation (red box: 558 μm, yellow box: 562 μm).

**Figure 2 bioengineering-12-01192-f002:**
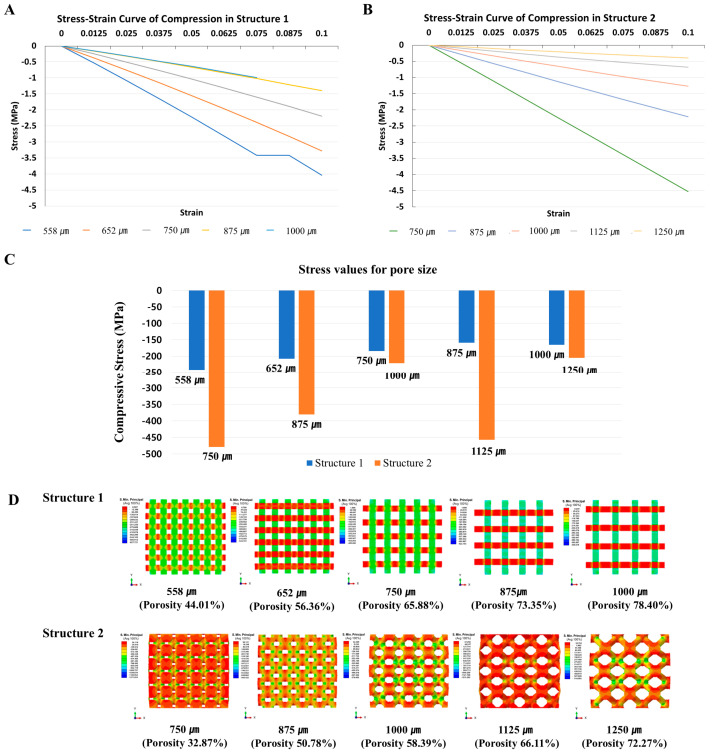
Biomechanical analysis of structure 1 and 2. (**A**,**B**). compressive strength derived from compression analysis. (**C**). Stress results derived from compression analysis of 10 types of scaffolds. (**D**). Stress distribution maps of 10 types of scaffolds derived from compression analysis.

**Figure 3 bioengineering-12-01192-f003:**
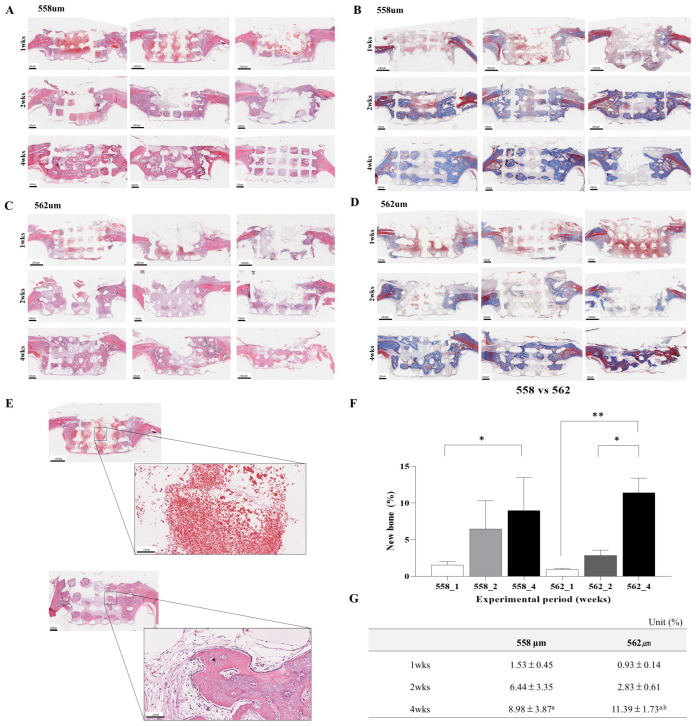
H&E and Masson’s Trichrome staining of the head and neck tissue in a rat implanted with 558 μm (**A**,**B**) and 562 μm (**C**,**D**) scaffolds. (**E**). H&E-stained image showing inflammatory tissue, osteoblasts, and osteoclasts (white arrows: osteoblasts; black arrows: osteoclasts). (**F**,**G**). Quantitative analysis of new bone formation in 558 μm scaffold and 562 μm scaffold in rats. * *p* < 0.05, ** *p* < 0.01. ^a^ Significantly different from 1 weeks (Statistical significance level was 5%, *p* < 0.05 and *p* < 0.01). ^b^ Significantly different from 2 weeks (Statistical significance level was 5%, *p* < 0.05).

**Figure 4 bioengineering-12-01192-f004:**
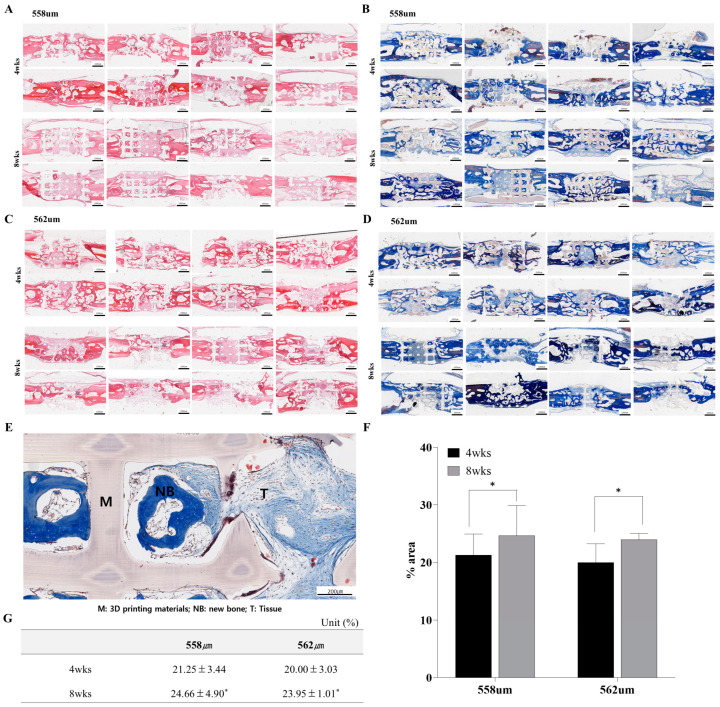
H&E and Masson’s Trichrome staining of the head and neck tissue in a rabbit implanted with 558 μm (**A**,**B**) and 562 μm (**C**,**D**) scaffolds. (**E**). Histological features between 3D printing materials. (**F**,**G**). Quantitative analysis of new bone formation in 558 μm scaffold and 562 μm scaffold in rabbits. * *p* < 0.01.

**Table 1 bioengineering-12-01192-t001:** Geometric Error Analysis between 3D CAD Model and Scaffold Printed Object.

Structure 1 (Lattice-Type)				Structure 2 (Dode-Type)			
Pore Size	Reference Pore Rate	Printed Object Pore Rate	Error	Pore Size	Reference Pore Rate	Printed Object Pore Rate	Error
558 μm	44.01%	49.01%	5.00%	750 μm	32.87%	33.01%	0.14%
652 μm	56.36%	53.30%	−3.06%	875 μm	50.78%	51.04%	0.26%
750 μm	65.88%	59.38%	−6.50%	1000 μm	58.39%	57.70%	−0.68%
875 μm	73.35%	73.33%	−0.01%	1125 μm	66.11%	66.07%	−0.04%
1000 μm	78.40%	76.59%	−1.81%	1250 μm	72.27%	65.53%	−6.74%
Average			3.28%	Average			1.57%

**Table 2 bioengineering-12-01192-t002:** Compression test results for scaffold material. Values are presented as mean ± standard deviation (SD).

Pore Size	Yield Displacement (mm)	Yield Load (N)	Yield Strain (%)	Yield Strength (MPa)	Elastic Modulus (MPa)
930 μm	0.98 ± 0.11	664.23 ± 80.47	3.85 ± 0.41	4.29 ± 0.50	164.87 ± 24.93
690 μm	1.02 ± 0.06	1107.06 ± 114.56 ^a^	3.78 ± 0.22	6.72 ± 0.68 ^a^	230.20 ± 25.08 ^a^
562 μm	0.84 ± 0.10	432.34 ± 53.23 ^b^	3.33 ± 0.38	2.80 ± 0.37 ^b^	144.22 ± 22.70 ^b^
558 μm	1.04 ± 0.11 *	582.05 ± 17.40 *^,a,b^	4.23 ± 0.45 *^,c^	4.02 ± 0.18 *^,a,b,c^	148.68 ± 12.10 *^,b^

^a^ Significantly different from 930 μm group in the same healing period (Statistical significance level was 5%, *p* < 0.05). ^b^ Significantly different from 690 μm group in the same healing period (Statistical significance level was 5%, *p* < 0.05). ^c^ Significantly different from 562 μm group in the same healing period (Statistical significance level was 5%, *p* < 0.05). * Significant different at the same observation period (Statistical significance level was 5%, *p* < 0.05).

**Table 3 bioengineering-12-01192-t003:** Fatigue test results of dental porous scaffolds.

Pore Size	Dynamic Compressive Load (N)	Number of Cycles	Test Result	Fracture Type
930 μm	#1: Min. = −30/Max. = −300 (R = 10)	5,000,000	Run out	
	#2: Min. = −70/Max. = −700 (R = 10)	5,000,000	Run out	
690 μm	#3: Min. = −30/Max. = −300 (R = 10)	5,000,000	Run out	
	#4: Min. = −70/Max. = −700 (R = 10)	5,000,000	Run out	
562 μm	#5: Min. = −30/Max. = −300 (R = 10)	5,000,000	Run out	
	#6: Min. = −70/Max. = −700 (R = 10)	10	Failure	Block fracture
558 μm	#7: Min. = −30/Max. = −300 (R = 10)	5,000,000	Run out	
	#8: Min. = −70/Max. = −700 (R = 10)	10	Failure	Block fracture

## Data Availability

The data are available in the tables in this manuscript. Other data used to support the findings of this study are available from the corresponding author upon request.

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
