# Peer review of "Optimized 3D-Printed Polylactic Acid/Graphene Oxide Scaffolds for Enhanced Bone Regeneration"

_bioengineering, 2025, doi:10.3390/bioengineering12111192_

Round 1

Reviewer 1 Report

Comments and Suggestions for Authors

1. The manuscript presents the development of 3D-printed PLA/Graphene oxide biocompatible scaffolds for bone regeneration. Lattice and dode type scaffolds were evaluated under compression and fatigue tests for dental applications and concluded that the dode structure exhibited superior performance.

2. In vivo animal study examined inflammatory responses in rats and bone regeneration in rabbits using two pore sizes (558 µm and 562 µm). After 8 weeks, no histological signs of inflammation were observed in rabbits. The 558µm group demonstrated enhanced bone growth of 24.66 +/- 4.90% within 8 weeks and greater resistance to premature fracture.

3. The scaffolds were fabricated using resin-based 3D printing, but essential printing parameters such as infill density, wall thickness, layer height, print speed, printing temperature, and nozzle diameter were not disclosed.

4. Mechanical tests were conducted under controlled environmental conditions to assess stress-strain characteristics. Compression tests were performed using a UTM at a loading rate of 1.3 mm/min with a 10% strain limit. Although the tests were stated to follow ASTM D696 standards, the reference citation is missing.

5. The fatigue test is reported to follow ASTM D695 standards. However, the test description involves tensile strength and negative Newton forces (which correspond to tension rather than compression). The authors should verify the standard as ASTM D638 may be more appropriate for tensile testing.

6. Fatigue testing was conducted using cyclic sinusoidal loading at 10 Hz. The rationale for selecting this specific frequency should be explained (particularly in relation to physiological relevance or prior studies).

7. The rat inflammation study involved two 5 mm circular calvarial defects, whereas the rabbit bone regeneration study used a single 5mm defect. The reason for this difference in experimental design should be clarified.

8. Statistical analysis was conducted using ANOVA with Tukey's post-hoc test, but an appropriate reference or justification for this method is missing.

9. At line 207, there is a table reference error (Table 3 is mentioned instead of Table 1 for porosity values).

10. In Section 3.1.1, the pore size of the printed scaffold is compared to the reference model. However, the methodology for3D printed pore measurement (e.g., microscopic imaging, vision-based measurement, or image analysis technique) should be clearly described.

11. At line 211, the reported shape error percentages show inconsistency. For example, Structure 1 exhibited 5% and -6.5% errors at 0.55 mm and 0.75 mm respectively, while the 0.65 mm structure produced -3.06%, higher than the 0.87mm and 1mm samples. Clarification is needed on how the average error was computed to ensure balanced interpretation.

12. In Figure 2C, the Y-axis scale and labels are not legible. Increasing the font size and improving figure clarity is recommended.

13. A reference for the Masson's Trichrome staining technique is missing. Including this will help readers better understand the histological evaluation of tissue components and staining characteristics.

Reviewer 2 Report

Comments and Suggestions for Authors

Round 2

Reviewer 2 Report

Comments and Suggestions for Authors

None